# Transformers over Directed Acyclic Graphs

**Yuankai Luo**
Beihang University
`luoyk@buaa.edu.cn`

**Veronika Thost**[*]
MIT-IBM Watson AI Lab, IBM Research
`veronika.thost@ibm.com`

**Lei Shi**[*]
Beihang University
`leishi@buaa.edu.cn`

## Abstract

Transformer models have recently gained popularity in graph representation learning as they have the potential to learn complex relationships beyond the ones captured by regular graph neural networks. The main research question is how to inject the structural bias of graphs into the transformer architecture, and several proposals have been made for undirected molecular graphs and, recently, also for larger network graphs. In this paper, we study transformers over directed acyclic graphs (DAGs) and propose architecture adaptations tailored to DAGs: (1) An attention mechanism that is considerably more efficient than the regular quadratic complexity of transformers and at the same time faithfully captures the DAG structure, and (2) a positional encoding of the DAG's partial order, complementing the former. We rigorously evaluate our approach over various types of tasks, ranging from classifying source code graphs to nodes in citation networks, and show that it is effective in two important aspects: in making graph transformers generally outperform graph neural networks tailored to DAGs and in improving SOTA graph transformer performance in terms of both quality and efficiency.

## 1 Introduction

Graph-structured data is ubiquitous in various disciplines [Gilmer et al., 2017, Zitnik et al., 2018, Sanchez-Gonzalez et al., 2020] and hence graph representation learning has the potential to provide huge impact. There are various types of *graph neural networks* (GNNs), the majority of which is based on a so-called message-passing scheme where node representations are computed iteratively by aggregating the embeddings of neighbor nodes [Gilmer et al., 2017]. Yet, this mechanism in its basic form has limited expressivity [Xu et al., 2018] and research is focusing on extensions.

*Transformer models* have recently gained popularity in graph learning as they have the potential to learn complex relationships beyond the ones captured by regular GNNs, in a different way [Dwivedi and Bresson, 2020, Kreuzer et al., 2021]. Technically, they can be considered as graph neural networks operating on fully-connected computational graphs, decoupled from the input graphs. The main research question in this context is how to inject the structural bias of the given input graphs (i.e., which nodes are actually connected) into the transformer architecture by adapting their attention and positional encoding appropriately. Several promising proposals have been made to encode undirected molecular graphs [Ying et al., 2021, Ross et al., 2022] and recent works take into account the scalability challenge [Dwivedi et al., 2022], also over larger network graphs [Rampášek et al., 2022, Chen et al., 2022b].

---

[*]Corresponding author.

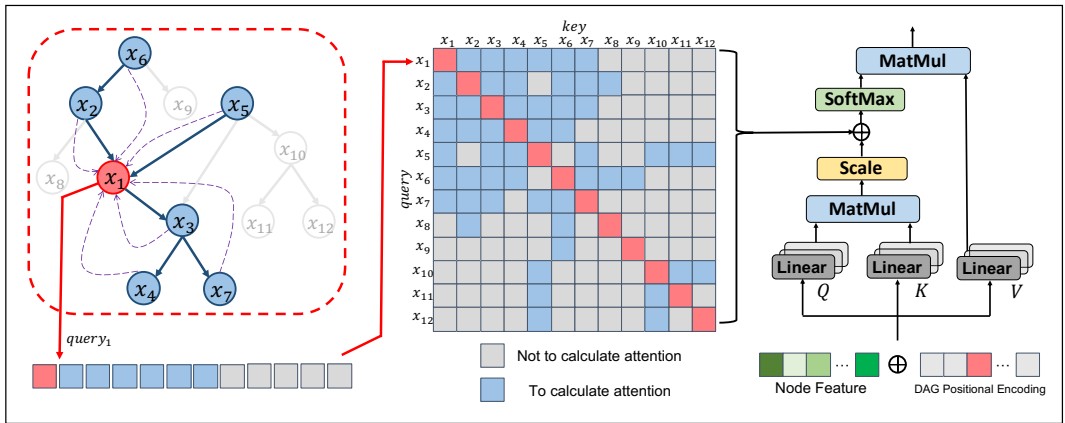

Figure 1: Overview of our DAG attention. A mask matrix restricts the receptive field of the node under consideration to nodes that are directly reachable in the DAG. Our positional encoding (right), additionally captures its position as its depth in the DAG.

We focus on *directed acyclic graphs* (DAGs), which are of special interest across various domains; examples include parsing results of source code [Allamanis et al., 2018], logical formulas [Crouse et al., 2019], and conversational emotion recognition [Shen et al., 2021], as well as probabilistic graphical models and neural architectures [Zhang et al., 2018, 2019, Knyazev et al., 2021]. In a DAG, the edges define a partial order over the nodes. This partial order represents an additional strong inductive bias, which offers itself to be integrated into the models.

In fact, various kinds of neural networks tailored to DAG structures have been proposed over the years; from early descriptions of recursive neural networks over DAGs [Sperduti and Starita, 1997, Frasconi et al., 1998], which have pointed out a similarity to sequence learning, to more recent works, such as DAG-RNN [Shuai et al., 2016], DAG-LSTM [Crouse et al., 2019], D-VAE [Zhang et al., 2019], and DAGNN [Thost and Chen, 2021]. The latter models focus on encoding the entire DAG; in a nutshell, they compute the embedding in a message-passing fashion by iterating over the DAG nodes in an asynchronous way, given by the partial order, and thereafter aggregating the final node representations into one for the DAG. This procedure yields state-of-the-art performance in terms of quality, but its asynchronous nature leads to comparatively (to regular GNNs) very slow runtime performance. The more parallel computation of transformers would seem to make them more suitable models and [Dong et al., 2022] have recently proposed one such model, PACE, which shows convincing performance in terms of both quality and efficiency.

In this paper, we focus on *transformers over DAGs* more generally, motivated by the highly flexible, parallel, and expressive nature of their computation. In particular, their success in sequence learning opens up the question how they can be tailored to DAGs, which are essentially sequential graphs. Based on the above-mentioned insights in DAG learning, we propose a straightforward and efficient **DAG attention framework**, which *effectively biases any transformer towards DAGs*.

- As outlined in Figure 1, we (1) adapt the attention mechanism by restricting the receptive field of each node to its predecessor and successor nodes so that it faithfully *captures the DAG structure - in terms of its reachability relation -* and at the same time gets *considerably more efficient* than the regular quadratic complexity; and (2) employ the positional encoding in a complementary way, to further bias the computation towards the DAG topology, by explicitly *encoding the node depth*.

- We show that the attention is not restricted effectively, even if it seems so technically. Moreover, we draw a connection to the random walk theory.

- We rigorously evaluate our proposal in ablation studies and show that it successfully improves different kinds of baseline transformers, from vanilla transformers [Vaswani et al., 2017] to state-of-the-art graph transformers [Wu et al., 2021, Chen et al., 2022a, Rampášek et al., 2022, Wu et al., 2022], over various types of DAG data. Our experiments range from classifying source code graphs to nodes in citation networks, and even go far beyond related works' problem scope. Most importantly, our proposal is proven *effective in two important aspects: in making graph*

*transformers generally outperform graph neural networks tailored to DAGs and in improving SOTA graph transformer performance in terms of both quality and efficiency.*

- Finally, *our DAG attention framework can be implemented in two possible and rather straightforward ways*, either on top of existing transformer models, or based on message-passing GNNs. Our implementation is available at https://github.com/LUOyk1999/DAGformer.

## 2 Background and Related Works

**Directed Acyclic Graphs.** We refer to a *graph* as tuple $G = (V, E, \mathbf{X})$, with node set $V$, edge set $E \subseteq V \times V$, and node features $\mathbf{X} \in \mathbb{R}^{n \times d}$, with each row representing the feature vector of one node, with number of nodes $n$ and feature dimension $d$, the features of node $v$ are denoted by $x_v \in \mathbb{R}^d$. A *directed acyclic graph* (DAG) is a directed graph without directed cycles. For a DAG $G$, we can define a unique *strong partial order* $\leqslant$ on the node set $V$, such that, for all pairs of nodes $u, v \in V$, $u \leqslant v$ if and only if there is a directed path from $u$ to $v$. We define the *reachability relation* $\leqslant$ over a DAG based on that partial order. That is, $u \leqslant v$ if and only if $v$ is reachable from $u$; further, if $u \leqslant v$, then $u$ is called a *predecessor* of $v$, and $v$ is a *successor* of $u$. All nodes without predecessors are *source* nodes, and all nodes without successors are *sink* nodes. We further define a restricted form of reachability $\leqslant_K$ with $u \leqslant_k v$ if and only if there is a directed path of length at most $k$ from $u$ to $v$. The *depth* of a node $v \in V$ is defined below. The *depth* of a DAG is the maximum depth over its nodes.

$$depth(v) = \begin{cases} 0 & \text{if } v \text{ is a source node} \\ 1 + \max_{(u,v) \in E} depth(u) & \text{otherwise} \end{cases}$$

**Message-Passing Graph Neural Networks.** Most modern *graph neural networks* (GNNs) are or can be formulated in terms of the *message-passing architecture*, a framework proposed in [Gilmer et al., 2017]. In that architecture, node representations are computed iteratively by aggregating the embeddings of their neighbor nodes (i.e., the messages), and a final graph representation can be obtained by aggregating the node embeddings. Yet, in its basic form, this mechanism has limited expressivity [Xu et al., 2018] and graph neural networks remain an active research area. Notable works we compare to include the **graph convolutional network (GCN)** [Kipf and Welling, 2017], a very early, basic architecture; the **graph attention network (GAT)** [Veličković et al., 2018], aggregating the neighbor node embeddings using attention; the **graph isomorphism network (GIN)** [Xu et al., 2018], which integrates extra MLP layers into message passing for increased expressivity; and the **principal neighbourhood aggregation (PNA)** model [Corso et al., 2020], a recent proposal focusing on adapting the aggregation mechanism to the node under consideration (e.g., scaling based on its degree). For a broader overview of the field, we refer the reader to [Wu et al., 2020].

**Transformers on Graphs.** Transformer models [Vaswani et al., 2017] have gained popularity in graph learning as they have the potential to learn complex relationships beyond the ones captured by regular GNNs and in a different way. The architecture is composed of two main parts: a *self-attention module* and a feed-forward neural network, each followed by a residual connection with a normalization layer. Formally, the self-attention projects the input node features $\mathbf{X}$ using three matrices $W_\mathbf{Q} \in \mathbb{R}^{d \times d_K}$, $W_\mathbf{K} \in \mathbb{R}^{d \times d_K}$ and $W_\mathbf{V} \in \mathbb{R}^{d \times d_K}$ to the corresponding representations for query ($\mathbf{Q}$), key ($\mathbf{K}$), and value ($\mathbf{V}$), and is described as follows :

$$\mathbf{Q} = \mathbf{X}W_\mathbf{Q}, \ \mathbf{K} = \mathbf{X}W_\mathbf{K}, \ \mathbf{V} = \mathbf{X}W_\mathbf{V},$$

$$\text{Attention}(\mathbf{X}) = \text{softmax}\left(\frac{\mathbf{Q}\mathbf{K}^T}{\sqrt{d_K}}\right)\mathbf{V}. \tag{1}$$

Over graphs, we focus on computing node instead of token embeddings (recall Figure 1). Technically, transformers can be considered as message-passing GNNs operating on fully-connected computational graphs, decoupled from the input graphs. The main research question in the context of graphs is how to inject the structural bias of the given input graphs by adapting their attention and by adding extensions to the original features, via so-called positional encodings (PEs). **Graph Transformer** [Dwivedi and Bresson, 2020] represents an early work using Laplacian eigenvectors as positional encodings, and various extensions and other models have been proposed since then [Min et al., 2022]. For instance, [Mialon et al., 2021] proposed a relative PE [Shaw et al., 2018] by means of kernels on graphs to bias the self-attention calculation. Notably, **GraphTrans** [Wu et al., 2021]

was the first hybrid architecture, using a stack of message-passing GNN layers before the regular transformer layers. Finally, [Chen et al., 2022a] have reformulated the self-attention mechanism as a kernel smoother as shown below and incorporated structure information into their **structure-aware transformer (SAT)** architecture by extracting a subgraph representation rooted at each node before attention computation:

$$\text{Attention}\,(x_v) = \sum_{u \in V} \frac{\kappa\,(x_v, x_u)}{\sum_{w \in V} \kappa\,(x_v, x_w)} f\,(x_u)\,, \forall v \in V, \tag{2}$$

with $f(x) = \mathbf{W_V} x$, and non-symmetric exp. kernel $\kappa$:

$$\kappa\,(x, x') = \exp\left(\frac{\langle \varphi(x)\mathbf{W_Q}, \varphi(x')\mathbf{W_K}\rangle}{\sqrt{d_K}}\right), \quad \varphi(x) = \begin{cases} x & \text{in vanilla transformer} \\ \text{GNN}_G(x) & \text{in SAT} \end{cases}$$

where $\langle \cdot, \cdot \rangle$ is the dot product on $\mathbb{R}^d$ and $\text{GNN}_G(x)$ is an arbitrary GNN model. Most of the aforementioned works focus on the classification of smaller graphs, such as molecules; yet, recently, **GraphGPS** [Rampášek et al., 2022] is also considering larger graphs and the area is focusing on the development of scalable models; for instance, **Nodeformer** [Wu et al., 2022] is designed to address the issue of scalability and expressivity in node classification. Altogether, the transformer architecture opens new and promising avenues for graph representation learning, beyond message-passing GNNs.

**Neural Networks over DAGs.** The additional strong inductive bias present in DAGs has motivated researchers to formulate special neural architectures tailored to this kind of graphs [Sperduti and Starita, 1997, Frasconi et al., 1998]. Various types of models have been proposed over the years in many different application areas. There are works in the context of syntactic graphs, [Tai et al., 2015, Shuai et al., 2016] logical formulas [Crouse et al., 2019], source code representations [Allamanis et al., 2018] and neural architectures [Zhang et al., 2018, 2019, Knyazev et al., 2021]. We particularly compare to the most recent proposals **S-VAE** and **D-VAE** [Zhang et al., 2019]; and to **DAGNN** from [Thost and Chen, 2021], who also showed that *using attention for neighbor node aggregation is beneficial in several DAG tasks*.

While the models can be formulated in terms of the message-passing framework [Thost and Chen, 2021], their processing is special for GNNs: they compute a graph embedding by iterating over the DAG nodes *in an asynchronous way*, given by the partial order, and starting from the source nodes (or the other way around, from the sink nodes). That is, the messages are passed along the partial order of the DAG and, at any point during computation, capture the information of the subgraph consisting of all the predecessors of the node currently under consideration. Thereafter, a final graph representations can be obtained by aggregating the embeddings of all sink nodes. In contrast to regular message-passing GNNs, these customized works for DAGs usually focus on encoding graphs (i.e., instead of nodes or edges, which are other common GNN targets) and pass messages over the entire graph, while regular GNNs consider a fixed radius - usually rather small - around each node and essentially obtain a neighborhood embedding. Furthermore, the nature of the proposed architectures shows that *recurrent techniques* have proven especially useful, *analogy to learning over sequences*. In summary, these DAG models are elegant and effective, yielding state-of-the-art results, but their asynchronous nature leads to very slow and practically inhibitive performance compared to regular GNNs.

**Transformers over DAGs.** We are aware of only few proposals for transformers tailored to DAGs. [Huang et al., 2022] developed a Directed Acyclic Transformer in the context of machine translation in order to simultaneously capture multiple translations within the decoded DAG, but the model's encoder is a vanilla transformer. [Kotnis et al., 2021] proposed BIQE, which leverages the depth of nodes in DAG paths as positional encodings, with a specific focus on answering complex queries in knowledge graphs. [Gagrani et al., 2022] proposed Topoformer, which is also an encoder-decoder model but has been developed for finding optimal topological orders in DAGs. Since the study entirely focuses on the latter goal (e.g., in terms of training objective and evaluation) it is very specific and different from our more general study. The DAG encoder itself is also rather different in that it uses a Laplacian PE, as it is used with regular graph transformers; and a more complex attention mechanism, which does not only model the reachability but also several other relations over the graph. Closest to our method is **PACE** [Dong et al., 2022], which similarly focuses on modeling the sequential nature of DAGs inside a transformer model and independently of a specific application in mind. However, (1) it applies a rather complex, node-specific positional encoding, whereas our PE only distinguishes

node depth; (2) the attention is based on the directed transitive closure, while we use reachability and show that this provides better quality; and (3) it's implemented using regular transformers, with runtime complexity $O(|V|^2 d)$, while we propose a much more efficient and scalable implementation based on message passing. Altogether, *our proposal is simpler and considerably more efficient. In addition, we do not propose a single model but a framework which can be flexibly applied on top of existing graph transformers and thus complement any custom, graph-specific transformer and adapt it to DAGs.*

## 3 Transformers for Directed Acyclic Graphs

Transformers have revolutionized deep learning, in particular sequence learning, and yield promising performance over graphs. Furthermore, we posit that their benefits actually match well the above-mentioned shortcomings of DAG neural networks. Therefore we developed a graph transformer framework tailored to DAGs. See Figure 1 for an overview.

### 3.1 Attention based on DAG Reachability

In contrast to regular graphs, the partial order in DAGs creates particular relations between connected nodes, in the sense that a given node's predecessors and successors are most likely more important to it than other graph nodes. Note that this intuition is also captured in the processing of DAG neural networks. Hence the reachability relation suggests itself to be exploited in our architecture. We apply it to restrict the receptive field of nodes during attention to their predecessors and successors in the graph. [Vaswani et al., 2017] already mentioned the possibility to use restricted attention in order to reduce complexity and, indeed, our proposal does not only yield an architecture which is biased towards the DAG structure, but additionally a considerably more efficient model.

While graphs to classify are usually of manageable size, graph learning in general may face much larger ones. For this reason, we formulate our model in a more general way, based on a bounded reachability relation, representing the receptive field of each node: $N_k(v) = \{(u, v) \in \leqslant_k\} \cup \{(v, u) \in \leqslant_k\}$. We adapt Equation (2) for our **reachability-based attention (DAGRA)**:

$$\text{Attention}_{\text{DAG}}(x_v) = \sum_{u \in N_k(v)} \frac{\kappa(x_v, x_u)}{\sum_{w \in N_k(v)} \kappa(x_v, x_w)} f(x_u), \forall v \in V.$$

The number $k$ represents a hyperparameter and can be chosen according to the data. In our ablation study (see Section 4), $k = \infty$ yielded best performance consistently. Observe that this choice of $k = \infty$ is still very different from both regular GNNs, which usually considerably restrict $k$, and regular transformers (Eq. 1), whose receptive field is not restricted at all (i.e., in terms of reachability).

### 3.2 Positional Encodings based on DAG Depth

As outlined in Section 2, positional encodings have been recognized as important and proven effective for incorporating graph structure into transformers. Observe that the sequential nature of DAGs makes them possess special position information, the depth of a node within the DAG. We propose to include this knowledge in the form of **directed acyclic graph positional encodings (DAGPE)** as follows, exactly as suggested for the original transformer architecture [Vaswani et al., 2017]:

$$PE_{(v,2i)} = \sin\left(\frac{depth(v)}{10000^{\frac{2i}{d}}}\right), \quad PE_{(v,2i+1)} = \cos\left(\frac{depth(v)}{10000^{\frac{2i}{d}}}\right),$$

where $d$ is the node feature dimension and $i$ the index of the dimension under consideration.

**DAG Attention.** We obtain the following model, combining DAGRA and DAGPE, for $v \in V$:

$$\text{Attention}(x_v) = \sum_{u \in N_k(v)} \frac{\kappa(x_v + PE_v, x_u + PE_u)}{\sum_{w \in N_k(v)} \kappa(x_v + PE_v, x_w + PE_w)} f(x_u). \tag{3}$$

Our attention incorporates both similarity of node features and of node depths. We argue that these are the most critical aspects of DAGs and our evaluation will show their impact and general effectiveness. In particular, note that most kinds of DAG data (e.g., citation networks) do not require us distinguishing between predecessor or successor nodes of same depth.

### 3.3 Expressive Power of DAG Attention

Technically, we restrict the attention to reachable nodes and, in this way, obtain considerable efficiency gains. Yet, our architecture is tailored to the special DAG structure, and we can show that this design offers similar expressivity to regular transformers.

It is important to note that in our framework, all nodes directly communicate with at least one source node (i.e., node without predecessors) by the DAG structure. This is specifically the case because we do not restrict the radius $k$ of the receptive field, but consider $k = \infty$. Hence, 2 layers are always enough to establish communication beyond any two nodes that have a common source node. Observe that this is often the case in practice; especially in DAG classification, many kinds of DAGs contain only a single source node (e.g., ogbg-code2 Hu et al. [2020] and NA Zhang et al. [2019]).

For DAGs with $m$ source nodes we need $2m$ layers for full communication, if we assume the DAG to be connected. In the latter case, every pair of source nodes has a common successor through which communication can happen. Further, connectedness is a reasonable assumption, otherwise communication is likely not needed in most scenarios.

### 3.4 Theoretical Intuition

We have shown that our architecture's bias emphasizes DAG relationships while re-directing the remaining relationships in the regular transformer's full attention matrix though the source nodes. This can also be shown to be in line with random walk theory, we draw a connection to PageRank [Brin, 1998, Gasteiger et al.]. Specifically, we consider a *PageRank variant that takes the root node into account - personalized PageRank.* We define the root node $x$ via the teleport vector $i_x$, which is a one-hot indicator vector. The personalized PageRank can be obtained for node $x$ using the recurrent equation $\pi_G(i_x) = (1-\alpha)A_{rw}\pi_G(i_x)+\alpha i_x$, with teleport (or restart) probability $\alpha \in (0, 1]$. Solving this equation, we obtain: $\pi_G(i_x) = \alpha(I_n - (1 - \alpha)A_{rw})^{-1}i_x$, where $A_{rw} = AD^{-1}$, with $A$ and $D$ being the adjacency and the degree matrix, respectively [Gasteiger et al.]. We invert the directions of the edges in $G$ to create a reverse DAG $\tilde{G}$. We can show that, for every node $x$, only nodes $y$ not reachable from $x$ (i.e., $y \notin N_\infty(x)$) will satisfy that $\pi_G(i_x)[y] + \pi_{\tilde{G}}(i_x)[y]$ (the y-th element of $\pi_{\tilde{G}}(i_x)$) equals 0. The proof is provided in Appendix A. This means that for the random walk's limit distribution, the probability of nodes that are not reachable from $x$ is 0.

### 3.5 Implementation

We describe two ways of implementing our model, especially DAGRA, based upon transformers and message-passing GNNs, respectively.

**Masked Attention for Transformers.** As shown in Figure 1, we can implement DAG attention in a very straightforward fashion using a mask that masks out node pairs based on the DAG reachability relation as follows (compare to Equation (1)), with the attention mask $\mathbf{M}$ being defined as a symmetric matrix over node pairs $(v, u) \in V \times V$.

$$\text{Attention}(\mathbf{X}) = \text{softmax}\left(\frac{\mathbf{Q}\mathbf{K}^T}{\sqrt{\text{d}_K}} + \mathbf{M}\right)\mathbf{V}, \quad \mathbf{M}(v, u) = \begin{cases} 0 & \text{if } u \in N_k(v) \\ -\infty & \text{otherwise.} \end{cases}$$

While this masking represents a simple technique to extend and bias existing transformer models to DAGs, the resulting architecture does not benefit from the restricted node set to be considered, leading to unnecessary, costly matrix operations during attention calculation. Moreover, it consumes $O(|V|^2)$ of additional memory to store $\mathbf{M}$. Thus, the runtime complexity per layer is still $O(|V|^2 d)$ for the vanilla transformer - and may be even higher, depending on the underlying transformer.

**DAG Attention using Message Passing.** Based on the formulation in Equation (3), we propose to follow the message-passing scheme; that is, for a node $v$, we compute $N_k(v)$ and only aggregate messages from the nodes in that set to compute the DAG attention. For the latter aggregation, we can use readily available frameworks, such as PyG [Fey and Lenssen, 2019]. We analyze the complexity of this proposal.

Table 1: Statistics of the datasets we used.

|  | ogbg-code2 | NA | Self-citation | Cora | Citeseer | Pubmed |
|---|---|---|---|---|---|---|
| # graphs | 452,741 | 19,020 | 1,000 | 1 | 1 | 1 |
| Avg # nodes | 125.2 | 8.0 | 59.1 | 2,708 | 3,327 | 19717 |
| Avg $n_\infty$ | 9.78 | 7.00 | 4.30 | 20.88 | 5.33 | 60.56 |

## 3.6 Computational Complexity

Clearly, the time complexity of the proposed model is lower than that of the standard transformer. We consider two computation steps:

**Computing DAG Reachability.** To obtain $N_k$, we compute the transitive closure of $E$ for each node in the graph using a breadth-first search starting at the node and iteratively expanding $N_k$ based on $E$. Hence, the overall complexity of this step is $O(|E||V|)$. Observe that, during training, we can consider this step as pre-processing since it only has to be run once, in the very beginning.

**DAG Attention.** The matrix product $\mathbf{Q}\mathbf{K}^T$ of self-attention has a cost $O(|V|^2 d)$ which is quadratic in the number of nodes in $G$, and hence especially critical for large graphs. Yet, for our DAG attention, we only aggregate messages from reachable nodes. Therefore, the time complexity of reachability relation attention is $O(|V| \times n_k \times d)$, where $n_k$ is the average size of $N_k$. In the worst case, we have $n_k = |V| - 1$, but we assume that $n_k << |V|$ in general. Especially for sparse graphs, the complexity gets significantly lower, i.e., $O(|V| \times \overline{deg}^k \times d)$, where $\overline{deg}$ is the average node degree.

Finally, observe that directed rooted trees represent a special kind of DAGs broadly seen across domains, such as abstract syntax trees of source code [Allamanis et al., 2018]. For them, the runtime of DAG attention scales almost linearly, as illustrated in the following theorem.

**Theorem 1.** *In a directed rooted tree $T$, the runtime of DAG attention is $O(|V| \times k \times \Delta^+ \times d)$, where $\Delta^+$ is the maximal outdegree of $T$. When $k = \infty$, the runtime of DAG attention is $O(|V| \times depth(T) \times \Delta^+ \times d)$.*

The proof is provided in Appendix A. Note that when $\Delta^+$ is small, $d$ is a constant and $depth(T)$ is $O(\log |V|)$, the runtime of DAG attention is almost linear in $|V|$. Indeed, we observed this on the ogbg-code2 dataset in our experiments.

# 4 Evaluation

We evaluate our proposed architecture on several datasets, comparing it to competitive baselines. Ablation studies also provide more insight into the composition of our model. Primarily, the following questions are investigated:

- Does **DAG attention** have the expected effects and **improves existing graph transformers** both in terms of quality and efficiency?
- Is DAG bias encoded through both **DAGRA & DAGPE**, and how does DAGPE compare to others?
- How does the **size of the receptive field** - the main difference between regular GNNs and transformers - affect the performance?

## 4.1 Experiment Setting

**Datasets.** Table 1 shows the diversity of the datasets we used; see Appendix B for full details.

- **ogbg-code2** [Hu et al., 2020]. A large dataset of ASTs derived from Python methods. The node features are syntax tokens. The multi-task classification task is to predict the first 5 tokens of the function name.
- **NA** [Zhang et al., 2019]. A dataset with much smaller graphs, containing neural architecture DAGs generated by the ENAS software. Each node's features represent a certain neural network component type. The (regression) task is to predict the architecture performance on CIFAR-10.
- **Self-citation** [ARC, 2021, Luo et al., 2023]. Each DAG in the academic self-citation represents a scholar's academic self-citation network [ARC, 2021]. Each paper has two node attributes: the

Table 2: **Code graph classification** on ogbg-code2. The baseline results were taken from the OGB leaderboard.

| Model | Valid F1 (%) | Test F1 (%) | Time(epoch) |
|---|---|---|---|
| GIN | $13.7 \pm 0.2$ | $14.9 \pm 0.2$ | 181s |
| GCN | $14.0 \pm 0.2$ | $15.1 \pm 0.2$ | 127s |
| GIN-Virtual | $14.4 \pm 0.3$ | $15.8 \pm 0.2$ | 155s |
| GCN-Virtual | $14.6 \pm 0.1$ | $16.0 \pm 0.2$ | 198s |
| GAT | $14.4 \pm 0.2$ | $15.7 \pm 0.2$ | 134s |
| PNA | $14.5 \pm 0.3$ | $15.7 \pm 0.3$ | 427s |
| DAGNN | $16.1 \pm 0.4$ | $17.5 \pm 0.5$ | 6018s |
| PACE | $16.3 \pm 0.3$ | $17.8 \pm 0.2$ | 2410s |
| Transformer | $15.5 \pm 0.2$ | $16.7 \pm 0.2$ | 1817s |
| **DAG+Transformer** | $\mathbf{17.4 \pm 0.1}$ | $\mathbf{18.8 \pm 0.2}$ | 591s |
| GraphTrans | $16.6 \pm 0.1$ | $18.3 \pm 0.2$ | 1117s |
| **DAG+GraphTrans** | $\mathbf{17.0 \pm 0.2}$ | $\mathbf{18.7 \pm 0.2}$ | 526s |
| GraphGPS | $17.4 \pm 0.2$ | $18.9 \pm 0.2$ | 1919s |
| **DAG+GraphGPS** | $\mathbf{17.6 \pm 0.1}$ | $\mathbf{19.3 \pm 0.2}$ | 608s |
| SAT (SOTA) | $17.7 \pm 0.2$ | $19.4 \pm 0.3$ | 2437s |
| **DAG+SAT** | $\mathbf{18.5 \pm 0.1}$ | $\mathbf{20.2 \pm 0.2}$ | 681s |

Table 3: **Node classification** results for the self-citation dataset; AP (%) and ROC-AUC (%).

| Model | AP ↑ | ROC-AUC ↑ |
|---|---|---|
| GIN | $57.7 \pm 1.8$ | $79.7 \pm 0.2$ |
| GCN | $58.8 \pm 0.4$ | $79.9 \pm 0.2$ |
| GIN-Virtual | $57.4 \pm 1.2$ | $79.5 \pm 0.4$ |
| GCN-Virtual | $58.9 \pm 0.2$ | $80.0 \pm 0.1$ |
| GAT | $55.3 \pm 3.7$ | $77.9 \pm 1.4$ |
| PNA | $62.4 \pm 0.7$ | $81.0 \pm 0.4$ |
| DAGNN | $61.2 \pm 0.6$ | $81.0 \pm 0.3$ |
| PACE | $52.1 \pm 1.8$ | $75.9 \pm 0.7$ |
| Transformer | $56.8 \pm 1.8$ | $78.7 \pm 0.3$ |
| **DAG+Transformer** | $\mathbf{63.8 \pm 0.8}$ | $\mathbf{82.2 \pm 0.5}$ |
| GraphGPS | $61.6 \pm 2.6$ | $\mathbf{81.3 \pm 0.6}$ |
| **DAG+GraphGPS** | $\mathbf{63.5 \pm 1.2}$ | $80.8 \pm 0.5$ |
| SAT | $59.8 \pm 1.7$ | $79.8 \pm 0.7$ |
| **DAG+SAT** | $\mathbf{62.7 \pm 1.5}$ | $\mathbf{80.6 \pm 0.7}$ |
| NodeFormer | $39.6 \pm 0.6$ | $69.4 \pm 0.3$ |
| **DAG+NodeFormer** | $\mathbf{64.9 \pm 0.8}$ | $\mathbf{81.7 \pm 0.8}$ |

publication year and total citation count (excluding the papers whose citation category is to be inferred). Here we consider the node-level task of predicting whether a paper is highly-cited or not - as a proxy for its impact.

- **Cora, Citeseer, Pubmed** [Sen et al., 2008]. Established, medium-sized citation graphs. Only for our method, we removed a small number of cyclic citation links to make them DAGs.

**Baselines.** We chose two basic message-passing GNNs, **GCN** and **GIN**; extensions of these models using a virtual node connected to all other graph nodes; the graph attention network **GAT**, as it showed especially good performance in [Thost and Chen, 2021]; **MixHop** [Abu-El-Haija et al., 2019], **LDS-GNN** [Franceschi et al., 2019], **IDGL** [Chen et al., 2020] and **PNA**, as more recent GNN proposals. In terms of transformer models, we considered vanilla **Transformer (TF)**, **Graph Transformer (GT)**, **GraphTrans**, **SAT**, **GraphGPS** and **NodeFormer** which achieved state-of-the-art performance (SOTA). Lastly, we consider neural networks tailored to DAGs: **S-VAE**, **D-VAE**, **DAGNN** and the current SOTA, **PACE**. For more detailed descriptions, see Section 2.

**DAG+ Models.** We implemented our DAG attention on top of vanilla Transformer, GraphTrans, SAT, GraphGPS and NodeFormer only (1) modifying their self-attention module by restricting attention in terms of reachability and (2) using DAGPE instead of the original one. We note that there are various alternatives for the latter (e.g., concatenation etc.), therefore we opted for the most simple solution which, at the same time, provides a very direct comparison. Moreover, on ogbg-code2 we did not use any PE since [Chen et al., 2022a] showed that they do not make real impact and we observed the same in preliminary experiments. For fair comparisons, we use the same hyperparameters (including the number of layers, batch size, hidden dimension etc.) and readout as baseline transformers. Given one of the baseline transformers M, we denote the modified model using DAG attention by **DAG+M**. Unless explicitly specified otherwise, we chose $k = \infty$ in all experiments. Full details on the experimental setup and hyperparameters are provided in the Appendix B.

### 4.2 Results and Discussion

**Overall Performance, Tables 2, 3, 4 and 5.** First, observe that the results are very consistent, although the datasets differ greatly in size, DAG sizes, DAG shape (e.g., in ogbg-code2 we have trees), and nature of data (e.g., node features). The message-passing GNNs represent standardized baselines but do not reach the performance of networks tailored to DAGs, such as DAGNN. Note that the latter is however comparatively bad on the node-level task. This can be explained by its processing. It computes node representations in the order of the partial order, and hence the representations of nodes in the beginning of the order contain only few information about the entire graph. Intuitively, transformers should be able to capture this information, but the transformers we tested do neither perform better, not even the ones tailored to graphs. This shows that they are missing information captured by those message-passing neural networks that were tailored to DAGs. The results clearly show that our DAG attention is successful in providing good improvements, over the best graph

Table 4: **Node classification accuracy** (%).The baseline results were taken from [Wu et al., 2022].

| Model | Cora | Citeseer | Pubmed |
|---|---|---|---|
| GCN | $87.06 \pm 0.34$ | $75.75 \pm 0.37$ | $88.16 \pm 0.14$ |
| GAT | $86.85 \pm 0.30$ | $75.92 \pm 0.26$ | $86.90 \pm 0.22$ |
| MixHop | $87.59 \pm 0.52$ | $73.64 \pm 0.73$ | $89.32 \pm 0.25$ |
| IDGL | $87.88 \pm 0.34$ | $74.32 \pm 0.51$ | $89.22 \pm 0.14$ |
| LDS-GNN | $87.82 \pm 0.62$ | $75.22 \pm 0.23$ | OOM |
| PACE | $79.47 \pm 0.63$ | $73.65 \pm 1.23$ | OOM |
| Transformer | $75.92 \pm 0.86$ | $72.23 \pm 1.06$ | OOM |
| **DAG+Transformer** | $\mathbf{87.80} \pm 0.53$ | $\mathbf{74.42} \pm 0.22$ | $\mathbf{89.01} \pm 0.13$ |
| SAT | $75.18 \pm 0.62$ | $74.88 \pm 0.73$ | OOM |
| **DAG+SAT** | $\mathbf{87.48} \pm 0.37$ | $\mathbf{76.64} \pm 0.26$ | $\mathbf{89.17} \pm 0.15$ |
| NodeFormer | $88.80 \pm 0.26$ | $76.33 \pm 0.59$ | $89.32 \pm 0.25$ |
| **DAG+NodeFormer** | $\mathbf{90.49} \pm 0.17$ | $\mathbf{78.24} \pm 0.33$ | $\mathbf{89.44} \pm 0.24$ |

Table 5: **Regression.** Predictive performance of latent representations over NA.

| Model | RMSE $\downarrow$ | Pearson's r $\uparrow$ |
|---|---|---|
| GCN | $0.482 \pm 0.003$ | $0.871 \pm 0.001$ |
| S-VAE | $0.521 \pm 0.002$ | $0.847 \pm 0.001$ |
| D-VAE | $0.375 \pm 0.003$ | $0.924 \pm 0.001$ |
| DAGNN | $0.264 \pm 0.004$ | $0.964 \pm 0.001$ |
| PACE | $0.254 \pm 0.002$ | $0.964 \pm 0.001$ |
| Transformer | $0.285 \pm 0.004$ | $0.957 \pm 0.001$ |
| GT | $0.275 \pm 0.003$ | $0.961 \pm 0.001$ |
| **DAG+Transformer** | $\mathbf{0.253} \pm 0.002$ | $\mathbf{0.966} \pm 0.001$ |
| GraphGPS | $0.306 \pm 0.004$ | $0.950 \pm 0.001$ |
| **DAG+GraphGPS** | $\mathbf{0.267} \pm 0.005$ | $\mathbf{0.964} \pm 0.001$ |
| SAT | $0.298 \pm 0.003$ | $0.952 \pm 0.001$ |
| **DAG+SAT** | $\mathbf{0.262} \pm 0.004$ | $\mathbf{0.964} \pm 0.001$ |

Table 6: Ablation results.

| Ablation | ogbg-code2 | | NA | | Self-citation | |
|---|---|---|---|---|---|---|
| | Valid F1 | Test F1 | RMSE $\downarrow$ | Pearson's r $\uparrow$ | AP $\uparrow$ | ROC-AUC $\uparrow$ |
| **DAG+TF** | $0.1731 \pm 0.0014$ | $0.1895 \pm 0.0014$ | $\mathbf{0.253} \pm 0.002$ | $\mathbf{0.966} \pm 0.001$ | $\mathbf{0.638} \pm 0.008$ | $\mathbf{0.822} \pm 0.005$ |
| (-) DAGRA | $0.1546 \pm 0.0018$ | $0.1670 \pm 0.0015$ | $0.284 \pm 0.003$ | $0.957 \pm 0.001$ | $0.573 \pm 0.011$ | $0.790 \pm 0.003$ |
| (-) DAGPE | $\mathbf{0.1739} \pm 0.0013$ | $\mathbf{0.1879} \pm 0.0015$ | $0.263 \pm 0.002$ | $0.963 \pm 0.001$ | $0.594 \pm 0.028$ | $0.782 \pm 0.018$ |
| (+) RWPE | - | - | $0.267 \pm 0.003$ | $0.962 \pm 0.001$ | $0.628 \pm 0.014$ | $0.819 \pm 0.010$ |
| (+) LapPE | - | - | $0.271 \pm 0.002$ | $0.961 \pm 0.001$ | $0.609 \pm 0.017$ | $0.786 \pm 0.015$ |
| (+) SPD Ying et al. [2021] | $0.1749 \pm 0.0011$ | $0.1881 \pm 0.0017$ | - | - | $0.639 \pm 0.006$ | $0.823 \pm 0.004$ |
| (+) Edge Direction | $0.1751 \pm 0.0018$ | $0.1870 \pm 0.0021$ | - | - | $0.636 \pm 0.015$ | $0.817 \pm 0.005$ |
| TF | $0.1546 \pm 0.0018$ | $0.1670 \pm 0.0015$ | $0.285 \pm 0.004$ | $0.957 \pm 0.001$ | $0.568 \pm 0.018$ | $0.787 \pm 0.003$ |
| **DAG+SAT** | $0.1821 \pm 0.0013$ | $0.1982 \pm 0.0010$ | $\mathbf{0.262} \pm 0.004$ | $\mathbf{0.964} \pm 0.001$ | $\mathbf{0.627} \pm 0.015$ | $\mathbf{0.806} \pm 0.007$ |
| (-) DAGRA | $0.1773 \pm 0.0023$ | $0.1937 \pm 0.0028$ | $0.292 \pm 0.003$ | $0.954 \pm 0.001$ | $0.598 \pm 0.031$ | $0.800 \pm 0.012$ |
| (-) DAGPE | $\mathbf{0.1846} \pm 0.0010$ | $\mathbf{0.2018} \pm 0.0021$ | $0.282 \pm 0.002$ | $0.958 \pm 0.001$ | $0.623 \pm 0.014$ | $0.806 \pm 0.005$ |
| (+) SPD Ying et al. [2021] | $0.1851 \pm 0.0008$ | $0.1991 \pm 0.0018$ | - | - | $0.627 \pm 0.016$ | $0.810 \pm 0.006$ |
| (+) Edge Direction | $0.1839 \pm 0.0014$ | $0.1978 \pm 0.0028$ | - | - | $0.623 \pm 0.013$ | $0.804 \pm 0.007$ |
| SAT | $0.1773 \pm 0.0023$ | $0.1937 \pm 0.0028$ | $0.298 \pm 0.003$ | $0.952 \pm 0.001$ | $0.598 \pm 0.017$ | $0.798 \pm 0.007$ |

transformer SAT and even better ones over vanilla Transformer. On ogbg-code2, the improvement is smaller for GraphTrans and SAT. However, this benchmark task is heavily dependent on other features (e.g., language understanding) and hence presents a special challenge; in this regard, the NA and self-citation datasets contain "cleaner" graphs. It is interesting to see that our framework lifts the transformers from below the level of DAGNN to above, in terms of all metrics, over these two datasets. Lastly, we observe that the PACE transformer tailored to DAGs is similarly outperformed by our simpler, but more effective technique. Nevertheless, the overall conclusion holds in general: Over all datasets, our DAG attention makes the transformers outperform (1) the original transformers and (2) the neural networks tailored to DAGs. This shows that the DAG-specific bias provided by DAG attention is the right bias. We investigated this in more detail in our ablation experiments.

**Ablation Study, Table 6.** Recall that our DAG attention is composed of the reachability-based attention (DAGRA) and DAGPE modules. To justify this design, (1) our ablation studies removing DAGRA and DAGPE individually confirm the impact of both modules; (2) we also experimented with replacing DAGPE with LapPE [Dwivedi and Bresson, 2020] and the random-walk-based RWPE [Dwivedi et al., 2021] to show that the DAG-specific nature of our PE is of advantage; (3) we experimented with adding attention bias which captures the graph structure more directly (e.g., shortest-path Ying et al. [2021] and edge directionality), although they are implicit in DAGPE; interestingly, the more direct representation does not make a noticeable difference. For ease of comparison and interpretation (i.e., in terms of magnitude), we also provide the baseline transformer results. Overall, it can be observed that our DAG attention yields highly consistent performance improvements, although occasionally the advantage is less pronounced. This shows that our architecture design provides the right bias on top of (graph) transformers, which makes the improved models better fit for DAGs.

**Impact of Size of Receptive Field $N_k$.**
**(1) Average $n_\infty$, Table 1.** Compared to the often only two to three hops considered in message-passing GNNs, our $k = \infty$ might seem unrealistic. However, as we show in the table, for two of our three and very different datasets, we have that $n_\infty$ is much smaller than the worst case size $|V|$. NA

Table 7: Impact of different $k$ on DAG attention, over ogbg-code2.

| $k$ | Avg $n_k$ | Test F1 score | |
| | | DAG+TF | DAG+SAT |
| --- | --- | --- | --- |
| 1 | 1.97 | $0.1724_{\pm 0.0022}$ | $0.1533_{\pm 0.0108}$ |
| 2 | 3.91 | $0.1800_{\pm 0.0023}$ | $0.1918_{\pm 0.0010}$ |
| 3 | 5.69 | $0.1842_{\pm 0.0006}$ | $0.1934_{\pm 0.0009}$ |
| 4 | 7.16 | $0.1849_{\pm 0.0021}$ | $0.1975_{\pm 0.0019}$ |
| 5 | 8.29 | $0.1856_{\pm 0.0020}$ | $0.2000_{\pm 0.0005}$ |
| 6 | 9.01 | $0.1869_{\pm 0.0020}$ | $0.2000_{\pm 0.0015}$ |
| 7 | 9.40 | $0.1869_{\pm 0.0019}$ | $0.2004_{\pm 0.0014}$ |
| 8 | 9.60 | $0.1875_{\pm 0.0008}$ | $0.2005_{\pm 0.0010}$ |
| $\infty$ | 9.78 | $0.1879_{\pm 0.0015}$ | $0.2018_{\pm 0.0021}$ |

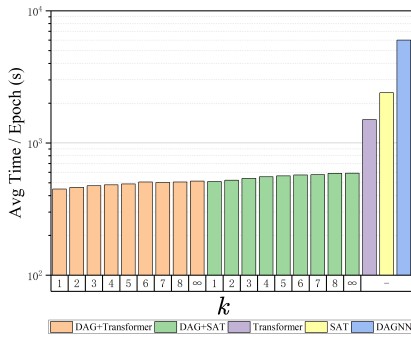

Figure 2: Average training time per epoch for various $k$ over ogbg-code2, log scale.

represents a special case because the graphs are very small so that we can always find a directed path through them that contains all nodes; however, given that $|V|$ is generally small, the exception of the rule $n_\infty << |V|$ is not critical here.

**(2) Performance for Varying $k$, Table 7, Figure 2.** We ran our model for different $k$ on ogbg-code2, based upon vanilla Transformer and SAT. We find that incorporating reachable node information leads to small but constant improvements in performance as $k$ increases. This confirms our intuition about the importance of predecessors and successors in DAGs, and is in line with related works suggesting to iteratively aggregate the DAGs along the partial order.

Figure 2 depicts the training time per epoch. It is easy to see that *our framework reduces computation time considerably*. For example, even the most time-consuming DAG+SAT requires 10 minutes per epoch, compared to 40 minutes for SAT and 100 minutes for DAGNN on the same GPU type. Since SAT and DAGNN represent the best-performing models among the transformers and, respectively, message-passing GNNs, and DAG attention yields qualitative improvements over both.

## 5   Conclusions

Based on the insights from graph neural networks and models for directed acyclic graphs, we have developed a transformer model biased towards DAGs, which allows for incorporating the main characteristic of DAGs - their partial order - into any transformer architecture, by encoding the reachability relation and positions resulting from it. Our architecture is simple and universal as we demonstrated in our experiments. Most importantly, it provides an effective extension improving existing graph transformers over DAGs. While our framework has successfully lifted existing (graph) transformers to the level of the state-of-the-art neural networks tailored to DAGs, our experiments show that there is still room for improvement. There is a variety of challenging tasks over many different kinds of DAGs, which are not fully solved yet; for instance, to tackle the challenge of reasoning over source code graphs, the models will probably need better language understanding. Hence DAG representation learning remains interesting for research, and our work provides an important contribution in closing the gap between transformers and other DAG neural networks.

**Limitations.** Our study is very general, but we found only a limited number of DAG datasets. We tried to resolve this by creating new ones (from the established citation datasets), yet we acknowledge that there is still room for extension. Further, there are likely specific types of DAGs which benefit from more customized modeling. Yet, our study is intended to address a general adaptivity to DAGs.

**Broader Impact.** Transformers have been popular in artificial intelligence, and it is expected that they also gain importance in graph learning. We hope to provide a tiny piece to the advancement of the area. Our framework is rather abstract, but its simplicity, efficiency, and universality make it practically usable; and we demonstrate good performance on a variety of data.

## Acknowledgments

We extend our gratitude to Yicheng Pan for his invaluable assistance with the computational complexity analysis. We also express our appreciation to all the anonymous reviewers for their insightful and constructive feedback. This work was supported by National Key R&D Program of China (2021YFB3500700), NSFC Grant 62172026, National Social Science Fund of China 22&ZD153, the Fundamental Research Funds for the Central Universities and SKLSDE.

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

# A Proofs

**Theoretical Intuition of DAG Attention**

*Proof.* Personalized PageRank is defined as $\pi_G(i_x) = (1 - \alpha)A_{rw}\pi_G(i_x) + \alpha i_x$. Through rearrangement, we obtain: $(I_n - (1 - \alpha)A_{rw})\pi_G(i_x) = \alpha i_x$. Now we prove that $(I_n - (1 - \alpha)A_{rw})$ is an invertible matrix. The matrix is invertible iff the determinant $\det(I_n - (1 - \alpha)A_{rw}) \neq 0$, which is the case iff $\det(A_{rw} - \frac{1}{1-\alpha}I_n) \neq 0$, i.e., iff $\frac{1}{1-\alpha}$ is not an eigenvalue of $A_{rw}$. This value is always larger than 1 since the teleport probability $\alpha \in (0, 1]$. However, the largest eigenvalue of the row-stochastic matrix $A_{rw}$ is 1, as can be proven using the Gershgorin circle theorem. Hence, $\frac{1}{1-\alpha}$ cannot be an eigenvalue and $(I_n - (1 - \alpha)A_{rw})$ is an invertible matrix. So we obtain $\pi_G(i_x) = \alpha(I_n - (1 - \alpha)A_{rw})^{-1}i_x$. Because the spectral radius (largest eigenvalue) of matrix $(1 - \alpha)A_{rw}$ is less than 1, from the Neumann series theorem, we obtain

$$(I_n - (1 - \alpha)A_{rw})^{-1} = \sum_{m=0}^{\infty} ((1 - \alpha)A_{rw})^m = \sum_{m=0}^{\infty} ((1 - \alpha)AD^{-1})^m.$$

Because $D$ is a diagonal matrix and $1 - \alpha$ is a constant, they do not affect whether an element in the matrix is 0 or not. Therefore, we only need to discuss $\sum_{m=0}^{\infty} A^m$. Because $A$ is the adjacency matrix of the DAG $G$, in this matrix, iff there is not a directed path from $x$ to $y$ ($x \nleq y$), we deduce that $(\sum_{m=0}^{\infty} A^m)[x, y] = 0$. This leads us to the conclusion that $\pi_G(i_x)[y] = 0$.

We invert the directions of the edges in $G$ to create a reverse DAG $\tilde{G}$. We can also obtain that, iff $x \nleq y$, then $\pi_{\tilde{G}}(i_x)[y] = 0$.

Therefore, for every node $x$, only nodes $y$ that $y \notin N_\infty(x)$ will satisfy that $\pi_G(i_x)[y] + \pi_{\tilde{G}}(i_x)[y]$ equals 0. □

**Theorem 2.** *In a directed rooted tree $T = (V, E)$ with the number of nodes $|V|$, the runtime of DAG attention is $O(|V| \times k \times \Delta^+ \times d)$, where $\Delta^+$ is the maximum out degree of the $T$ and $d$ is feature dimension. When $k = \infty$, the runtime of DAG attention is $O(|V| \times \ell \times \Delta^+ \times d)$, where $\ell$ is the depth of $T$.*

*Proof.* In DAG attention, for each node $v$, we need to calculate the attention to $N_k(v)$, whose runtime is upper bounded by $O(|N_k(v)| \times d)$. So the runtime of DAG attention is linear in the sum of the sizes of receptive fields $\sum_{v \in V} |N_k(v)|$. We only need to show that $\sum_{v \in V} |N_k(v)| \leqslant |V| \times k \times \Delta^+$. When $k = \infty$, $\sum_{v \in V} |N_k(v)| \leqslant |V| \times \ell \times \Delta^+$.

Next, we prove this theorem for the case of $k = \infty$. For bounded $k$, the proof is similar. We prove by induction on the depth $\ell$ of $T$.

We assume that $\Delta^+ \geqslant 2$, since when $\Delta^+ = 1$, $T$ is a 1-ary tree, $\sum_{v \in V} |N_\infty(v)| = |V| \times |V| \leqslant |V| \times \ell \times \Delta^+$.

For the basis step, when $\ell = 0$, $T$ only has one node. So $\sum_{v \in V} |N_\infty(v)| = 0 \leqslant |V| \times \ell \times \Delta^+$.

For the inductive step, we assume that $\sum_{v \in V} |N_\infty(v)| \leqslant |V| \times \ell \times \Delta^+$ holds when $\ell \geqslant 0$. Consider a tree $T_{new}$ of depth $\ell + 1$ whose root is denoted by $r$. Then the sub-trees whose roots are the children of $r$ have depth at most $\ell$. Let $s$ be the number of these sub-trees, $s \leqslant \Delta^+$, and $V_i$ be the node set of the $i$-th sub-tree. Since the number of reachability relations with $r$ involved is $2(|V| - 1)$, we have

$$
\begin{aligned}
&\sum_{v \in V} |N_\infty(v)| \\
=\ & \sum_{i \in [s]} \sum_{v \in V_i} |N_\infty(v)| + 2(|V| - 1) \\
\leqslant\ & |V_1| \times \ell \times \Delta^+ + ... + |V_s| \times \ell \times \Delta^+ + 2(|V| - 1) \\
\leqslant\ & \ell \times \Delta^+ \times (|V| - 1) + 2(|V| - 1) \\
\leqslant\ & \ell \times \Delta^+ \times (|V| - 1) + \Delta^+ \times (|V| - 1) \\
\leqslant\ & (\ell + 1) \times \Delta^+ \times |V|
\end{aligned}
$$

Table 8: The statistics of academic self-citation dataset.

|            | Min | Avg | Max |
|------------|-----|-----|-----|
| # nodes    | 30  | 59  | 296 |
| DAG depth  | 1   | 5   | 21  |
| Node degree| 0   | 1.8 | 83  |

This completes the proof of Theorem 2 for the case of $k = \infty$. $\qquad\qquad\square$

# B  Experimental Details

## B.1  Computing Environment

Our implementation is based on PyG [Fey and Lenssen, 2019]. All reported results are averaged over 10 runs using different random seeds. The experiments are conducted with two RTX 3090 GPUs.

## B.2  Dataset Details

We provide details of the datasets used in our experiments, including ogbg-code2 [Hu et al., 2020], Neural architectures (NA) [Zhang et al., 2019], Self-citation, Cora, Citeseer and Pubmed [Sen et al., 2008].

**ogbg-code2.** ogbg-code2 [Hu et al., 2020] is a dataset containing 452,741 Python method definitions extracted from thousands of popular Github repositories. It is made up of Abstract Syntax Trees as DAGs where the task is to correctly classify the sub-tokens that comprise the method name. The min/avg/max numbers of nodes in the graphs are 11/125/36123, respectively. We use the standard evaluation metric (F1 score) and splits provided by [Hu et al., 2020].

**Neural architectures (NA).** This dataset is created in the context of neural architecture search. It contains 19,020 neural architectures generated from the ENAS software [Pham et al., 2018]. Each neural architecture has 6 layers (i.e., nodes) sampled from 6 different types of components, plus an input and output layer. The input node vectors are one-hot encodings of the component types. The weight-sharing accuracy [Pham et al., 2018] (a proxy of the true accuracy) on CIFAR-10 [Krizhevsky et al., 2009] is taken as performance measure. Since it is a regression task, the metrics are RMSE and Pearson's r. We use the splits as is used in [Thost and Chen, 2021]. Details about the generation process can be found in [Zhang et al., 2019].

**Self-citation.** The academic self-citation dataset is a directed acyclic graph, representing the scholar's academic self-citation network on the natural language processing (NLP) field extracted from ACL Anthology Reference Corpus [ARC, 2021]. This dataset contains 1000 academic self-citation networks from scholars ranked top-1000 by number of papers. In a self-citation networks, each node is a paper of this scholar and each directed edge indicates that one paper is cited by another one. Each graph node includes two quantitative attributes: the publication year, the paper's total citation count. The task is to predict missing citation counts given existing citation counts. Specifically, for each scholar self-citation network, we randomly select 10% of its papers and drop their citation counts, and the node-level task is to predict whether a paper which misses its citation count is highly-cited or not. In Computer Science, Engineering, and Mathematics, for papers in ISI (Web of science database) listed journals that are 10 years old, around 20 citations gets in the top 10%. So we consider papers with a citation count greater than or equal to 20 as highly-cited papers. As the class balance is skewed (only 27.2% of data is positive), we use the Average Precision (AP) and ROC-AUC as the evaluation metrics. We randomly split the dataset into 80% training, 10% valid and 10% test sets. The statistics is shown in Table 8.

**Cora, Citeseer and Pubmed** [Sen et al., 2008]. These three datasets are commonly used citation networks for evaluating models on node classification tasks. These datasets are medium-scale networks (with 2K~20K nodes) and the goal is to classify the topics of documents (instances) based on input features of each instance (bag-of-words representation of documents) and graph structure (citation links). For the baselines, the citation links are treated as undirected edges [Wu et al., 2022] and each document has a class label. Only for our method, we treated citation links as directed edges and removed a small number of cyclic citation links to make them DAGs. We use the same evaluation metric (classification accuracy) and splits provided by NodeFormer [Wu et al., 2022].

Table 9: Hyperparameter search on different datasets.

| Hyperparameter | NA | Self-citation | Cora, Citeseer and Pubmed |
|---|---|---|---|
| # Layers | {2,3} | {2, 3, 4, 5} | {2, 3, 4, 5} |
| Hidden dimensions | {32,128} | {32, 64, 128} | {16, 32, 64, 128} |
| Dropout | 0.2 | {0.1, 0.2, 0.5} | 0.0 |
| Learning rate | 1e-3 | {1e-4, 5-e4, 1e-3, 5e-3} | {1e-3, 1e-2} |
| # Epochs | 100 | {50, 100} | 1000 |
| Weight decay | None | 1e-6 | 5e-3 |
| # Attention heads | {2, 4} | {2, 4, 8} | {2, 4} |

Table 10: Number of parameters for DAG+ models.

| Model | ogbg-code2 | NA | Self-citation | Cora | Citeseer | Pubmed |
|---|---|---|---|---|---|---|
| DAG+Transformer | 127,06k | 399k | 478k | 291k | 404k | 14k |
| DAG+SAT | 149,53k | 631k | 710k | 276k | 165k | 14k |

## B.3 Hyperparameters and Reproducibility

**ogbg-code2.** We implemented DAG attention on vanilla Transformer, GraphTrans, GraphGPS and SAT that we only modify the self-attention module to DAG attention module. And we do not use any PE as well as baselines because it makes very little performance improvement [Chen et al., 2022a]. For fair comparisons, we use the same hyperparameters (including training schedule, optimizer, number of layers, batch size, hidden dimension etc.) as baseline models for all of our four versions.
**NA.** We train the transformer-based baselines (vanilla Transformer, GraphGPS and SAT) in front of DAGNN [Thost and Chen, 2021]. For SAT, we use the 3-subtree GNN extractor (GIN). For GraphGPS, we use GatedGCN as GPS-MPNN and vanilla Transformer as GPS-GlobAttn. And we implement DAG attention on those respectively. We follow the exact training settings of DAGNN [Thost and Chen, 2021]. Then we also train a sparse Gaussian process (SGP) [Snelson and Ghahramani, 2005] with a learning rate of 4e-4, a epochs number 200 as the predictive model to evaluate the performance of the latent representations. The transformer-based hyperparameters are summarized in Table 9. All other hyperparameters are the same as those of DAGNN [Thost and Chen, 2021].
**Self-citation.** We implement DAG attention on the vanilla Transformer, GraphGPS, SAT and NodeFormer. In general, we perform a hyperparameter search to produce the results for our model and baselines. The hyperparameters on the academic self-citation dataset are summarized in Table 9, where # Attention heads is tuned for transformer-based models. For SAT, we conduct positional encodings search on LapPE-8, RWPE-20 or None, and we use the 3-subtree GNN extractor (GIN). For GraphGPS, we use GatedGCN as GPS-MPNN and vanilla Transformer as GPS-GlobAttn. The other hyper-parameters for NodeFormer are the default parameters in their public code. In all experiments, we use the validation set to select the best hyperparameters. All our models and baselines are trained with the AdamW optimizer [Loshchilov and Hutter, 2016] and we use the cross-entropy loss on this classification task. The learning rate scheduler is the cosine scheduler [Loshchilov and Hutter, 2016].
**Cora, Citeseer and Pubmed**. We implement DAG attention on the vanilla Transformer, SAT and NodeFormer. Only for Pubmed, we choose the size of the receptive field $k = 2$ (otherwise $k = \infty$). And we perform a hyperparameter search to produce the results for our model and those transformer-based baselines in Table 9. We follow all other training setting and hyperparameters of NodeFormer [Wu et al., 2022]. For SAT, we use the 1-subtree GNN extractor (GCN).

In Table 10, we report the number of parameters for DAG+ models using the hyperparameters selected from Table 9.

## C Model Visualization

Here, we demonstrate that the DAG transformer also provides interpretability compared to the traditional Transformer model. We trained a DAG+Transformer and a Transformer model on a self-citation graph, and compared the attention scores between the selected node and other nodes

learned by both models. Figure 3 visualizes the results. In the leftmost graph, the darker blue of the nodes represents a higher number of citations. For the selected node $v$, DAG attention only puts attention weights on the nodes that have citation relations with it ($N_k(v)$), whereas classic self-attention focuses on nodes that are almost unrelated to $v$. More focus on these core nodes makes the DAG model less influenced by other patterns and thus leads to improved performance.

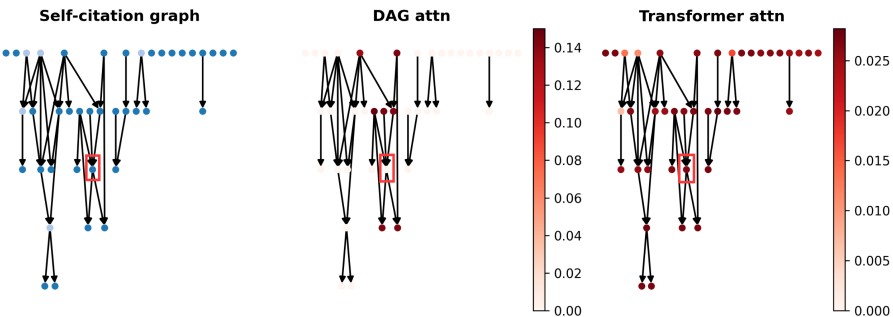

Figure 3: Attention visualization of DAG+Transformer and the Transformer on self-citation. The figure shows the attention weights of the node in the red box learned by DAG attention and the classic self-attention.

# D   Additional Results

## D.1   Semi-supervised Node Classification

We test our model on three citation networks Cora, Citeseer and Pubmed. We implement DAG attention on DIFFormer-a Wu et al. [2023]. Table 11 reports the testing accuracy. This show that our framework likely increases both quality and runtime here as well.

Table 11: **Node classification accuracy** (%).The baseline results were taken from [Wu et al., 2023].

| Model | Cora | | Citeseer | | Pubmed | |
|---|---|---|---|---|---|---|
| | Accuracy | Time (s) | Accuracy | Time (s) | Accuracy | Time (s) |
| NodeFormer | $83.4_{\pm 0.2}$ | - | $73.0_{\pm 0.3}$ | - | $81.5_{\pm 0.4}$ | - |
| DIFFORMER-a | $84.1_{\pm 0.6}$ | 14.21 | $75.7_{\pm 0.3}$ | 8.48 | $80.5_{\pm 1.2}$ | 1013.31 |
| DAG + DIFFORMER-a | $85.1_{\pm 0.7}$ | 10.96 | $76.2_{\pm 0.4}$ | 7.01 | $81.6_{\pm 0.5}$ | 84.15 |

## D.2   MalNet-Tiny

Here we consider MalNet-Tiny. MalNet-Tiny [Freitas and Dong, 2021] is a subset of MalNet which consists of function call graphs (FCGs) obtained from Android APKs. This subset includes 5,000 graphs with a maximum of 5,000 nodes, originating from benign software or 4 malware types. The FCGs do not have any original node or edge features. The benchmark version of this dataset usually employs Local Degree Profile as the set of node features. And the objective is to predict the type of software solely based on the structure.

Note that we did not do any hyperparameter tuning. Further, we implemented DAG attention on top of GraphGPS by only modifying the self-attention module, switching it to DAG attention. As shown in Table 12, Our DAG+GraphGPS achieves a test accuracy of 93.45% while only requiring 20 seconds per epoch. This demonstrates the quality and efficiency of DAG attention.

# E   Further Related Works

**Neural Networks on Directed Graphs.** Most works we found focus on adapting the Laplacian: [Ma et al., 2019] construct a directed Laplacian matrix to improve the propagation model by using identities that involve the random walk matrix and its stationary distribution. [Monti et al., 2018] consider several different symmetric Laplacian matrices corresponding to local directed graph motifs.

Table 12: **Graph classification** on MalNet-Tiny.

| Model | Accuracy (%) | Time(epoch) |
|---|---|---|
| GraphGPS | $92.64 \pm 0.78$ | 46s |
| **DAG+GraphGPS** | $93.45 \pm 0.41$ | 20s |

Moreover, DiGCN [Tong et al., 2020] is based on a directed Laplacian with a PageRank matrix to learn multi-scale features and [Zhang et al., 2021] proposed MagNet, a GNN for directed graphs based on the Magnetic Laplacian. The latter has been recently integrated into Transformers in the form of a positional encoding [Geisler et al., 2023]. Note that this improves predictive performance in directed graphs, but does impact the (quadratic) complexity of self-attention.

**Transformers over Trees.** Observe that trees are special in that they are a particular, more restricted type of DAGs. [Sun et al., 2020] proposed TreeGen, a tree-based Transformer that combines grammar rules and tree structure using a tree reader. However, it is not directly relevant to our work as it focuses on program-specific information. In addition, there are some methods of leveraging tree paths and integrating them into the self-attention module. [Alon et al., 2018] used AST to represent a source code snippet. They considered pairwise paths between leaf nodes, representing them as sequences of the leaf and non-leaf nodes in AST. [Kim et al., 2021] used another type of path that goes from the leaf nodes up to the AST root by traversing its ancestors. [Zügner et al.] presented Code Transformer, which combines distances computed on ASTs and context of source code in the self-attention. Specifically, it utilizes four different paths to reason about ASTs relative positions. Furthermore, [Peng et al., 2022] designed a new directed tree Transformer position encoding for each node based on a two-dimensional description of tree structures. In contrast to these works, we shift our focus toward more general DAGs, use a very simple and intuitive PE, and also adapt the receptive field of the attention.

