# OpenReview forum: "Transformers over Directed Acyclic Graphs"
_NeurIPS.cc/2023/Conference — NeurIPS 2023 poster_

### Official Review · Reviewer_xj5i · 2023-06-29

**Soundness:** 2 fair
**Presentation:** 2 fair
**Contribution:** 1 poor
**Rating:** 4
**Confidence:** 4

**Summary:**

The paper proposes an attention mechanism for directed acyclic graphs. Specifically the reachability to/from nodes is considered for a given node and the unreachable nodes are masked out. The authors use this attention mechanism in existing graph transformers for undirected graphs. Further the paper also proposes a new PE for directed graphs that considers the maximum depth of the node from the root nodes. The method is evaluated over source code and citation datasets and authors show it improves existing transformer performance.

**Strengths:**

1. The method achieves state of the art results when induced in transformer architectures on datasets used
2. Method is claimed to be parallelizable and results in faster training compared to asynchronous methods


**Weaknesses:**

1. Regarding limiting the receptive field of the transformer, the method selects nodes up to path length k. But a principle question arises about “How to select k?”. As of now it seems to be a heuristic or a dataset dependent parameter, which limits the novelty of the work as it is straightforward to apply attention in the top k hop reachable nodes in order to induce some notion of structure in the transformer.
2. Considering that the method only limits the attention to reachable nodes and uses topological depth as PE in existing methods the novelty and contribution seems limited.
3. The proposed attention mechanism that is restricted to reachable nodes, may suffer from learning issues when a node label depends on non-reachable nodes. For example if the label of a child node depends on sibling nodes but those are not reachable and the transformer used is of single layer. The same issue may arise with more layers etc. if there exists label dependence on farther non-reachable nodes. This is where the fundamental strength of the transformer lies over GNN that it doesn't need to wait over many layers for the information to be propagated. But with the proposed modification this property is lost.

Missing Citations:
1. https://proceedings.mlr.press/v162/dong22b.html


**Questions:**

1. Regarding limiting the receptive field of the transformer, the method selects nodes up to path length k. But a principle question arises about “How to select k?”. Can the authors comment on a more principled way of selecting k rather than keeping it a heuristic or a dataset dependent parameter?
2. In table 6, the results for the base transformer model and proposed method without DAGRA are the same in ogbg-code2. Is this because no PE is used and the rest of the architecture remains the same?
3. Do the authors see any issues in learning with the proposed attention mechanism that is restricted to reachable nodes? For example if the label of a child node depends on sibling nodes but those are not reachable and the transformer used is of single layer. The same issue may arise with more layers etc. if there exists label dependence on non-reachable nodes. This is where the fundamental strength of the transformer lies over GNN that it doesn't need to wait over many layers for the information to be propagated. But with the proposed modification this property is lost. Can the proposed framework be adapted to handle such cases?


**Limitations:**

Limitations of the method have been addressed in the paper.

---

> ### Author Rebuttal · Authors · 2023-08-09
>
> **Thank you for the constructive comments!**
>
> We indeed missed to discuss the rather important Q3 in the paper and detail the actual power of our DAG attention in the global reply. There we also add details about the comparison to the reference mentioned. We are sorry for the confusion caused by the missing explanation and hope that our contributions are clearer now.
>
> Please let us know if there are any questions left!
>
>
> **Q1 Clarification: Receptive Field**
>
> Note that we generally recommend $k=\infty$ (line 273), which means we consider all nodes related via the reachability relationship. This is a reasonable choice given our theoretical analysis, based on the proven random walks, and has also shown best performance in all our experiments. In fact, observe that our framework distinguishes itself in that it is easy to use, requiring no hyperparameter selection or tuning (see also WzDZ, Q1).
>
> **Q2 Setting over ogbg-code2**
>
> Yes, in ogbg-code2, no PE is used, the rest of the architecture remains the same. Similar to the transformer baselines we used, we did not see improvement with PE; see also lines 268-270 in the paper. We hypothesize that this is related to the underlying transformer and also the data, since there is a work using a PE [1], but they consider textual information beyond the graph recommended in the benchmark.
>
> [1] Zügner et al. "Language-Agnostic Representation Learning of Source Code from Structure and Context." ICLR 2020.

---

> > ### Comment · Reviewer_xj5i · 2023-08-13
> > **Response to Rebuttal**
> >
> > Thanks to the authors for the clarifications and it helps my understanding of the proposed method and contributions. I will think through the discussed points in detail and consider my review in light of the responses.
> >
> > Many Thanks

---

> ### Author Response · Authors · 2023-08-13
> **Response by Authors**
>
> Thank you for getting back to us and for considering our rebuttal in the final review!

---

> ### Author Response · Authors · 2023-08-21
> **Author Question**
>
> Dear Reviewer xj5i,
>
> We hope the clarifications in the rebuttal changed your view of our contribution to the positive side.
>
> We are sorry for bothering you again! But since the competition is tough and you acknowledged our rebuttal, we wanted to make sure our work doesn't get forgotten.

---

> > ### Comment · Reviewer_xj5i · 2023-08-21
> >
> > I have read the rebuttal and I want to express my appreciation for the response by authors. The response basically address my questions. Therefore, I will keep my rating unchanged so far and make the final decision until the reviewer discussion phase. Many Thanks!

---

> ### Author Response · Authors · 2023-08-21
> **Final Comment by Authors**
>
> Thank you so much for your participation in the discussion and for mentioning it to us!

---

### Official Review · Reviewer_WzDZ · 2023-06-30

**Soundness:** 4 excellent
**Presentation:** 3 good
**Contribution:** 3 good
**Rating:** 6
**Confidence:** 4

**Summary:**

This paper adapts transformers to directed acyclic graphs. It restricts the receptive field of each node to its predecessor and successor nodes so that it faithfully captures the DAG structure. It also incorporates positional encodings based on the node depth. Extensive experiments show that it can improve performance of different kinds of baseline transformers.

**Strengths:**

1. The proposed method is efficient and versatile. It can improve performance of a broad spectrum of transformers.
2. The problem of adapting transformers to DAGs is under-explored.
3. The analysis and experimental results regarding $k$ is surprising and inspiring.

**Weaknesses:**

1. The datasets used in this paper are rather small. As efficiency and versatility are two key advantages claimed by the paper, can the authors include experimental results on larger graphs, like ogbn datasets?

**Questions:**

1. The authors mentioned that they used the same hyperparameters as baseline transformers. Does this include learning rate, weight decay and dropout? If so, one only needs to integrate the proposed module without searching any hyperparameters (as long as $k$ is large enough) and can immediately facilitate an enhancement in performance.
2. See weakness 1.


**Limitations:**

Most limitations have been addressed.

---

> ### Author Rebuttal · Authors · 2023-08-09
>
> **Thank you for the thoughtful comments!**
>
> These are interesting points that highlight our contibution, and we included them into the paper.
>
> **Q1 No Hyperparameter Tuning**
>
> We indeed used the same hyperparameters as the baseline transformers, which include learning rate, weight decay, and dropout. Extensive experiments showed that integrating our proposed module without the need for hyperparameter search can lead to an immediate improvement in performance, and hence considerably ease usability.
>
> **Q2 Larger Datasets**
>
> So far, we focused on datasets which were considered with models tailored to DAGs or transformers in the past. Due to the limited time in the rebuttal period, we report only some numbers here, but we will conduct further experiments and also consider other transformers.
>
> **ogbn-arxiv, Citation Data.** We removed a small number of cyclic citation links to create DAGs, used the official splits, averaged over 5 runs, and compared vanilla Transformer to similarly proven GNNs. The results demonstrate that we can adapt the transformers to large graphs and achieve competitive results.
>
>
>    | Methods    | GCN              | GraphSAGE        | GPRGNN           | Transformer | DAG+Transformer  |
>    | ---------- | ---------------- | ---------------- | ---------------- | ----------- | ---------------- |
>    | ogbn-arxiv | 71.72 $\pm$ 0.45 | 71.46 $\pm$ 0.26 | 70.90 $\pm$ 0.23 | OOM         | 71.53 $\pm$ 0.34 |
>
>
> **MalNet-Tiny, Function Call Graphs (FCGs)[1].** 5k graphs with max. 5k nodes, originating from benign software or 4 malware types, these 5 types are to be predicted. We use the very same setting as GraphGPS, which employs the Local Degree Profile as the set of node features, and added DAG attention on top of GraphGPS by only modifying the self-attention module, switching it to DAG attention. Note that we did not do any hyperparameter tuning. DAG+GraphGPS achieves a slightly better test accuracy while only requiring 20 seconds per epoch. This confirms the quality and efficiency of DAG attention.
>
> | Model            | Accuracy (%)         | Time(epoch) |
> | ---------------- | -------------------- | ----------- |
> | GraphGPS         | 92.64 $\pm$ 0.78     | 46s         |
> | **DAG+GraphGPS** | **93.45 $\pm$ 0.41** | **20s**     |
>
> -----------------------------------
>    [1] Freitas, et al. "A large-scale database for graph representation learning." NeurIPS 2021

---

> > ### Comment · Reviewer_WzDZ · 2023-08-13
> > **Response to rebuttal**
> >
> > Thanks for the rebuttal, most of my concerns have been addressed.

---

> > > ### Comment · Reviewer_WzDZ · 2023-08-13
> > > **One more suggestion**
> > >
> > > Let me give another suggestion to the related works section: The paper could have benefited from acknowledging [1,2,3] as pertinent references and conducting a comparative analysis with the proposed method. This would have contributed to a more comprehensive examination of the relevant works and underscored the innovative nature of the proposed methodology.
> > >
> > >
> > > [1] Chengxuan Ying, Tianle Cai, Shengjie Luo, Shuxin Zheng, Guolin Ke, Di He, Yanming Shen, and Tie-Yan Liu. Do transformers really perform bad for graph representation? In Advances in Neural Information Processing Systems, 2021.
> > >
> > > [2] Md. Shamim Hussain, Mohammed J. Zaki, and Dharmashankar Subramanian. Global self attention as a replacement for graph convolution. In ACM SIGKDD Conference on Knowledge Discovery and Data Mining, 2022.
> > >
> > > [3] Qitian Wu, Chenxiao Yang, Wentao Zhao, Yixuan He, David Wipf, and Junchi Yan. 2023. DIFFormer: Scalable (Graph) Transformers Induced by Energy Constrained Diffusion. In International Conference on Learning Representations, 2023.

---

> ### Author Response · Authors · 2023-08-13
> **Thank you for initiating further interesting discussion!**
>
> **It is a good idea to point out that the focus of SOTA research [2,3] on regular graph transformers is very different from our proposal**, which shows that we can exploit the special structure of DAGs to considerably improve their effectiveness in this very specific - but sometimes important - setting.
>
> We can certainly add the indeed proven method [1] as baseline. Please also note that we experimented with the shortest path distance from that paper to address U6X8, Q3.
>
> Please note that, since there are many interesting SOTA graph transformers, our submission focused on GraphGPS and NodeFormer, which provided SOTA scores at the time when we were conducting our experiments. We'll run further experiments to include [3] and  [3] + DAG into the tables now. **This could indeed nicely underline the fact that our framework's general nature allows for continuous and even subtle SOTA advances** (i.e., here, comparing NodeFormer and DIFFormer [3]). We'll update you here once the results are available!

---

> > ### Comment · Reviewer_WzDZ · 2023-08-14
> > **Response to rebuttal**
> >
> > Allow me to clarify my previous point. My intention was simply to recommend incorporating references to these works within the related works section, thereby enhancing the comprehensiveness of the background. I understand that the rebuttal timeframe is limited, and my intention was not to burden you with additional efforts in introducing more baselines or applying the proposed method to new backbones. After all, you have already applied the proposed method to 3 backbones and I think it is enough. Nevertheless, in case circumstances permit, you could consider including the previously mentioned experimental outcomes in the final version. Thank you for the feedback.

---

> ### Author Response · Authors · 2023-08-14
> **Thank you for clarifying!**
>
> We wanted to make sure we address *all* remaining concerns and the experiment is a valid proposal. Preliminary experiments on some datasets show that **our framework likely increases both quality and runtime here as well, sometimes considerably**.
>
> |                   | Cora               |                | Citeseer           |                |
> | :---------------- | :----------------- | :------------- | :----------------- | :------------- |
> |                   | Accuracy           | Train time (s) | Accuracy           | Train time (s) |
> | NodeFormer        | 83.4 $\pm$ 0.2     | -              | 73.0 $\pm$ 0.3     | -              |
> | DIFFormer-a       | 84.1 $\pm$ 0.6     | 14.215         |75.7 $\pm$ 0.3     | 8.481          |
> | DAG+DIFFormer-a | **85.1** $\pm$ 0.7 | **10.967**     | **76.2** $\pm$ 0.4 | **7.015**      |
>
> Please note that we report the semi-supervised setting (different from our paper), since this is the one considered in the DIFFormer paper.
>
> (edit: we found a bug in the pre-processing with Citeseer and adapted those results)

---

> > ### Comment · Reviewer_WzDZ · 2023-08-15
> > **Response to rebuttal**
> >
> > Thanks for the additional results, which contribute to further substantiating the versatility of the proposed method. I have raised my rating to 6. However, I do have a minor question: why does the inclusion of DAG can save the training time?

---

> ### Author Response · Authors · 2023-08-15
> **This is an important point**
>
> Thank you for asking!
>
> Since we technically (though not effectively, see the global reply) restrict the attention to only the DAG's relationships, which are are usually sparse, and implement this using message-passing GNNs, **we obtain considerable performance increases generally, also during training**.
>
> More precisely, the time complexity is reduced to $O(|V | × n_\infty × d)$; $n_\infty$ denotes the average size of the receptive field and is typically much smaller than |V|. See also Sec. 3.5.

---

### Official Review · Reviewer_Qc5n · 2023-07-06

**Soundness:** 3 good
**Presentation:** 3 good
**Contribution:** 3 good
**Rating:** 6
**Confidence:** 3

**Summary:**

The paper proposed a new approach for DAG representation learning using transformer. The representation learning on DGA is significant as DAG can be adapted into many real-world problems, which is also explained in the paper. In addition, the DAG can be formed into a sequence of nodes so that it is naturally to think about using transformers to model. The paper conducted comprehensive experiments to support its claim.

**Strengths:**

The paper utilizes transformer to achieve representation learning on DAG, which is natural. Although transformer has been used on graphs, they are mostly for undirected graphs.

The paper tells the story in an easy-to-read way. It also conducts extensive experiments to show the superior performance of the proposed model.

The task the paper wants to solve is significant in that DAG can be naturally adapted to many real-world applications. This has also been discussed in the apper.

**Weaknesses:**

(1) It seems that there is one paper that does the same thing [1]. I think this paper should at least be discussed in the paper.
(2) I was wondering why some models in Table 2 and Table 2 are not present in Table 4 and Table 5, such as PNA, DGANN.
(3) Will the choice of the root node affect the final results?

[1] Dong et al., PACE: A Parallelizable Computation Encoder for Directed Acyclic Graphs. 2022

**Questions:**

See weakness.

**Limitations:**

Limitations have been discussed by the paper.

---

> ### Author Rebuttal · Authors · 2023-08-09
>
> **Thank you for the constructive comments!**
>
>
> We indeed missed the PACE paper, please see the global reply for a detailed discussion and experimental results. Overall, the comparison to this transformer tailored to DAGs highlights the effectiveness of our simpler and more general proposal: PACE is already outperformed by vanilla Transformer+DAG.
>
> We also clarified the below points in the paper.
>
> **Q2 Choice of Baselines**
>
> Since we used well-known datasets, we mostly resorted to the models usually reported with those, or in other DAG-specific papers. We did not run those by ourselves, but are happy to add some if you have recommendations. DAGNN specifically is hard to be run - if it runs at all - on the larger datasets since it is very expensive (e.g., it takes days for the ogbg-code2 experiments). This is also why we (and also the PACE paper) put such emphasis onto efficiency.
>
> **Q3 Root Nodes are Given**
>
> The term "root node" refers to the source nodes, i.e., nodes without predecessors. These are fixed, that is, they are given with the datset and do not represent architecture choices or parameters. Hence they do not affect the results.

---

> > ### Comment · Reviewer_Qc5n · 2023-08-14
> > **Response to author**
> >
> > Thank the author for clarifying my concerns.
> >
> > a. I still think that it's a bit weired if baselines in Table 2, 4 and 5 are different. Although this is a minor issue, I would recommend the author to make it equivalent for fair comparison and statement in the paper.
> >
> > b. For the source node, how is it specified in the dataset, by random? I would also suggest the author to clarify this in their paper.

---

> ### Author Response · Authors · 2023-08-14
> **Response by Authors**
>
> a. Please note that we did not intend to ignore the comparison, but rather did not want to give the impression that is reasonable to run DAGNN over larger graphs, especially in case where there are alternatives; observe that it is more than 100 times slower than GCN or our model. Nevertheless, this experiment reveals that our framework, even in combination with the most simple vanilla Transformer, provides a useful alternative and is **likely able to make any transformer competitive to SOTA networks tailored to DAGs**.
>
> |       | Cora             | Citeseer         | Pubmed           | NA                |                   |
> | ----- | ---------------- | ---------------- | ---------------- | ----------------- | ----------------- |
> |       | Accuracy         | Accuracy         | Accuracy         | RMSE              | Pearson's r       |
> | PNA   | 87.03 $\pm$ 0.73 | 73.11 $\pm$ 1.06 | 87.99 $\pm$ 0.26 | 0.691 $\pm$ 0.003 | 0.707 $\pm$ 0.001 |
> | DAGNN | 84.49 $\pm$ 0.59 | **74.52 $\pm$ 0.67** | OOM              | 0.264 $\pm$ 0.004 | 0.964 $\pm$ 0.001 |
> |Transformer |75.92 $\pm$ 0.86| 72.23 $\pm$ 1.06 |OOM|0.285 $\pm$0.004   |0.957 $\pm$ 0.001  |
> |**DAG+Transformer**| **87.80** $\pm$ 0.53| **74.42 $\pm$ 0.22** |**89.0** $\pm$ 0.13|**0.253** $\pm$ 0.002   |**0.966** $\pm$0.001  |
>
> b. Sorry for not specifying this clearly, there is no randomness here and we use the same setting as the related works. More specifically, every dataset comes with the graphs in terms of nodes and edges and, by definition (and without any randomness), source nodes are the nodes where the indegree is zero. For instance, in python source code graphs, this might be nodes corresponding to a code token in the very beginning (e.g., "def") and, in citation networks, this are nodes corresponding to authors who cite others but who have not been cited yet.

---

> > ### Comment · Reviewer_Qc5n · 2023-08-15
> > **Response to author**
> >
> > Thank the response from the author and my concerns are properly addressed.

---

> ### Author Response · Authors · 2023-08-15
> **Thank you so much!**
>
> This final confirmation is very helpful and highly appreciated.

---

### Official Review · Reviewer_U6X8 · 2023-07-07

**Soundness:** 3 good
**Presentation:** 3 good
**Contribution:** 3 good
**Rating:** 6
**Confidence:** 3

**Summary:**

The paper proposing a new Transformer-based graph neural network for directed graphs (DAGs) that restricts the receptive field size of self-attention and adds depth-based node embeddings to improve learning from DAGs. The resulted model is more efficient than previous Graph Transformer models and at the same time more effective in different DAG tasks as shown in many experiments.

**Strengths:**

1. The problem of learning from DAGs is interesting and important. Improving Transformers for this problem is a reasonable direction.
2. The introduced improvements are simple, but effective.
3. Computational complexity analysis in Section 3.5 is a nice addition.
4. Experiments are overall convincing.
5. The paper is easy to follow and presented well.

**Weaknesses:**

While I enjoyed reading the paper, the paper needs to address the following weaknesses:

1. The paper is missing related works [A, B], which seem to be doing exactly the same as DAGNN by proposing what they call GatedGNN. Therefore, citing DAGNN but not citing A/B can be misleading for readers.

2. Papers [C, D] are also related and it would be nice to discuss them as well. Given their recent appearance it's not necessary to empirically compare to them (although it would make the paper stronger), but they should be discussed. In particular, [C] is solving the same ogbg-code2 task, but achieves much better performance, can the authors discuss why? [D] is proposing a Transformer-based [40] graph model for DAGs and also add node depth-based PE embeddings similar to those in this submission + degree-based embeddings based on [40] (which could further improve the results in this submission). Given these papers, L122: "to the best of our knowledge, transformers have not been studied particularly in the context of DAGs" should be rephrased.

3. The DAGRA component of the method does not seem to leverage two important sources of information: (1) the direction of edges -- node predecessors and successors are treated equally based on L163-164; (2) the shortest path distance between nodes -- the mask is equal to 0 or −∞ in L203.  Even though the DAGPE embeddings can help to recover this information, leveraging (1) and (2) seems very natural and could improve results by better distinguishing certain DAG structures. In related papers using Transformers for undirected/directed graphs, (1) and (2) are leveraged. For example, in [C] the forward and backward edges are treated separately, and in [40], [C], the shortest path distance is used in masking.

3. Section 3.3's conclusion is not very clear. Is this section trying to say that all nodes will eventually communicate with each other if enough layers and/or large k are used? Please elaborate/rephrase.

4. Results on ogbg-code2 in Table 6 are a bit confusing, because at first it looks like DAG+TF/SAT are using DAGPE, but from the text it sounds like DAGPE is not used. I think it would be more clear to report both results, with DAGPE and without it, even if the latter is better so that the Table does not have blank entries.

**Minor issues:**

- "partial order" - wouldn't it be more appropriate to use "topological order" that is often used in DAG context (e.g. https://networkx.org/documentation/stable/reference/algorithms/generated/networkx.algorithms.dag.topological_sort.html and [A, B])? "partial order" seems to be more relevant to sets.

-  In Tables, bolding results per pair makes sense, but baseline's top performance should be highlighted too. Also, the overall top performance across all rows can be underlined (or highlighted in some other way).

- In Fig. 3, it's hard to conclude that DAG+Nodeformer separate classes better, so I don't think this visualization is helpful in the main text.

- All equations should be numbered so that reviewers and readers can refer to them easily.

- L130: "in more important fields than" - it's a quite subjective statement
- L154: "[a] given node’s predecessors"
- L169: "regular transformers (e.g., Eq. 2)" - Eq. 1?
- L168: "still very different" - please elaborate how is it different given that k=∞ is the best

*References:*

- [A] Graph HyperNetworks for Neural Architecture Search. ICLR 2019.
- [B] Parameter Prediction for Unseen Deep Architectures. NeurIPS 2021.
- [C] Can We Scale Transformers to Predict Parameters of Diverse ImageNet Models? ICML 2023.
- [D] Transformers Meet Directed Graphs. ICML 2023.

**Questions:**

See Weaknesses above.

I'm looking forward to the author's response and will be willing to update my score.

**Limitations:**

Limitations are discussed, which is appreciated. L344 says that "we found only a limited number of DAG datasets", so one suggestion would be to use a dataset of DAGs introduced in [B].

---

> ### Author Rebuttal · Authors · 2023-08-09
>
> **Thank you for the very detailed feedback!**
>
> Q4 raises an important point which we address in the global reply. We also added some of the discussion below to the paper since it covers interesting aspects. For results on additional datasets, please see the reply to WzDZ, Q2. We sincerely thank you for giving us the opptunity to improve the score and are happy to answer further questions if needed.
>
>
> **Q1 Related Works**
>
> Thank you for sharing these related works. We added discussion about them to the paper.
>
> **Q2 Recent Related Works**
>
> We have noticed a potential mix-up in the citations of [C] and [D] since "[C] Can We Scale Transformers ..." does not address the ogbg-code2 task. We found [D] on arxiv in spring and checked out the code, yet:
> - They did not use the data as it is suggested by the OGB benchmark but adapted the graph construction (e.g., making it more succinct by removing redundancy). This likely also addresses the question why they reach such good performance in parts.
> - They did not employ the OGB evaluator.
> - They do not provide the new datasets they constructed on GitHub, making a proper comparison with their work challenging.
>
> For these reasons, we refrained from directly comparing to [D] since this seems not straightforward and beyond the focus of our study.
>
> We updated L122 and also the related works section.
>
> **Q3 On Incorporating Graph Structure Differently**
>
> We intended to design our framework in the most simple and efficient way, in order to ease usability, and therefore decided to focus on the main characteristics of DAGs.
>
> While there may be features that can improve our framework in certain scenarios, we will likely never obtain an exhaustive model; several other graph features have been shown powerful (e.g., sibling distances [1]) and others will be shown useful in the future. Nevertheless, when applying our model to [40], we consider the shortest path distance, and applying it to [C], we can take into account the direction of edges. And of course, we can combine them if needed in a specific context.
>
> That said, it is an interesting question, in how far such features actually increase performance, since they are implicitely covered by the PE. The below results over ogbg-code2 and self-citation seem to suggest that our simple DAG PE is surprisingly strong, as the impact of (1) edge directionality and (2) shortest path distances is limited here.
>
> |                         | ogbg-code2              |                         | self-citation         |                       |
> | ----------------------- | ----------------------- | ----------------------- | --------------------- | --------------------- |
> |                         | Valid F1 score          | Test F1 score           | AP                    | ROC-AUC               |
> | DAG+transformer         | **0.1739 $\pm$ 0.0013** | **0.1879 $\pm$ 0.0015** | **0.638 $\pm$ 0.008** | **0.822 $\pm$ 0.005** |
> | DAG+transformer+(1)     | 0.1751 $\pm$ 0.0018     | 0.1870 $\pm$ 0.0021     | 0.636 $\pm$ 0.015     | 0.817 $\pm$ 0.005     |
> | DAG+transformer+(2)     | 0.1749 $\pm$ 0.0011     | 0.1881 $\pm$ 0.0017     | 0.639 $\pm$ 0.006     | 0.823 $\pm$ 0.004     |
> | DAG+transformer+(1)+(2) | 0.1750 $\pm$ 0.0017     | 0.1884 $\pm$ 0.0012     | 0.637 $\pm$ 0.005     | 0.821 $\pm$ 0.004     |
> | DAG+SAT                 | **0.1846 $\pm$ 0.0010** | **0.2018 $\pm$ 0.0021** | **0.627 $\pm$ 0.015** | **0.806 $\pm$ 0.007** |
> | DAG+SAT+(1)             | 0.1839 $\pm$ 0.0014     | 0.1978 $\pm$ 0.0028     | 0.623 $\pm$ 0.013     | 0.804 $\pm$ 0.007     |
> | DAG+SAT+(2)             | 0.1851 $\pm$ 0.0008     | 0.1991 $\pm$ 0.0018     | 0.627 $\pm$ 0.016     | 0.810 $\pm$ 0.006     |
> | DAG+SAT+(1)+(2)         | 0.1852 $\pm$ 0.0013     | 0.1986 $\pm$ 0.0019     | 0.628 $\pm$ 0.007     | 0.811 $\pm$ 0.005     |
>
>
>
> **Q5 Additional ogbg-code2 Results**
>
> This is a fair point, we revised the table as follows.
>
>
>    |           | Valid F1 score      | Test F1 score       |
>    | --------- | ------------------- | ------------------- |
>    | DAG+TF    | 0.1731 $\pm$ 0.0014 | 0.1895 $\pm$ 0.0014 |
>    | (-) DAGPE | 0.1739 $\pm$ 0.0013 | 0.1879 $\pm$ 0.0015 |
>    | DAG+SAT   | 0.1821 $\pm$ 0.0013 | 0.1982 $\pm$ 0.0010 |
>    | (-) DAGPE | 0.1846 $\pm$ 0.0010 | 0.2018 $\pm$ 0.0021 |
>
> **Q6 Minor Issues**
>
> Thank you for checking on that level of detail! We updated the paper accordingly.
>
> Regarding $k=\infty$, note that this setting still strongly decreases the complexity compared to regular transformers, since only the reachable nodes are considered for attention. However, given the special DAG structure, this likely does not decrease the effective attention. A detailed discussion on the power of our attention is given in the global reply.
>
> ----------------------
> [1] Zügner et al. “Language-Agnostic Representation Learning of Source Code from Structure and Context.” ICLR 2020.

---

> > ### Comment · Reviewer_U6X8 · 2023-08-14
> >
> > Thank you for the response. It clarifies some of my concerns, therefore I increased the score from 5 to 6.

---

> ### Author Response · Authors · 2023-08-14
> **Thank you for getting back to us!**
>
> We highly appreciate the careful and positive evaluation.
>
> In case there is anything we can do to address the remaining concerns, please let us know!

---

### Author Rebuttal · Authors · 2023-08-09

**We thank all reviewers for the very fair, detailed, and constructive feedback!**

We are sorry for the unnecessary confusion caused by missing **explanation about the power of our DAG attention** and hope that the below details clarify our contribution. Since the scores are borderline overall, we would appreciate if the reviewers consider that. We are happy to provide additional information if needed.

We also provide a **detailed comparison to a very related paper we missed, but which overall highlights the potential impact** of our work.

------------

**G1 Power of DAG Attention**

Technically, we restrict the attention to reachable nodes and, in this way, obtain considerable efficiency gains. Yet, our architecture is tailored to the special DAG structure, and we can show that this design offers similar expressivity to regular transformers.

It is important to note that in our framework *all* nodes directly communicate with at least one source node (i.e., node without predecessors) by the DAG structure. This is specifically the case because we set $k=\infty$. Hence, *2 layers are always enough to establish communication beyond any two nodes that have a common source node*. Furthermore, especially DAG classification datasets usually contain DAGs with a single source node (e.g., ogbg-code2 and NA).

For *DAGs with $m$ source nodes we need $2m$ layers for full communication*, if we assume the DAG to be connected. In the latter case, every pair of source nodes has a common successor through which communication can happen. Further, connectedness is a reasonable assumption, otherwise communication is likely not needed in most scenarios. Our empirical results demonstrate that our design likely does not limit expressivity on many, also large datasets.

The source nodes may seem to represent a certain kind of bottlenecks since, essentially, our architecture's bias emphasizes DAG relationships while re-directing the remaining relationships in the regular Transformer's full attention matrix. But Sec. 3.3 shows that the *importance of relationships we model is in line with well-known random walk theory*, and our qualitative performance increases also demonstrate that putting more emphasis on DAG relationships can be beneficial in a variety of use cases.

---------------

**G2 Comparison to PACE [1]**

As two reviewers noted, we missed a very related transformer architecture tailored to DAGs, PACE, published at ICML 2022. Interestingly, the comparison serves well to highlight our contribution:
- Most importantly, *PACE is one specific model* whereas we propose a framework which can be flexibly used with any transformer.
- Specifically, *PACE uses a more complex PE* and its attention is *based only on the directed transitive closure*. In contrast, we use reachability, which also accounts for data where reverse relationships are critical.
- PACE injectively maps DAGs to sequences of node embeddings and then processes them as a sequence in a masked transformer. The mask restricts the attention, with an intention similar to our restricted attention. Yet, *the model still has the usual quadratic complexity* of transformers, while we propose an implementation based on message passing GNNs that exploits our novel attention to reduce complexity.
- Further, PACE is limited in applicability, as the masked transformer *fails to consider edge attributes* of graphs, while our framework preserves all graph information.
- Our framework's simple design also seems to be more effective empirically: *even in combination with vanilla Transformer, we obtain better results*. Given the simplicity of our model, we especially see considerable efficiency gains.
    |                 | ogbg-code2       |                  |             | self-citation    |                   | Cora               | Citeseer            | Pubmed             |
    | --------------- | ---------------- | ---------------- | ----------- | ---------------- | ----------------- | ------------------ | ------------------- | ------------------ |
    |                 | Valid F1 score   | Test F1 score    | Time(epoch) | AP               | ROC-AUC           | Accuracy           | Accuracy            | Accuracy           |
    | PACE            | 16.3$\pm$0.3     | 17.8$\pm$0.2     | 2410s       | 52.1$\pm$1.8     | 75.9$\pm$0.7      | 79.47$\pm$0.63     | 73.65$\pm$1.23      | OOM                |
    | DAG+Transformer | **17.4$\pm$0.1** | **18.8$\pm$0.2** | **591s**    | **63.8$\pm$0.8** | **82.2$\pm$75.9** | **87.80$\pm$0.53** | **74.42$\pm$0.22** | **89.01$\pm$0.13** |

[1] PACE: A parallelizable computation encoder for directed acyclic graphs, ICML 2022.

------------------------------

**G3 Novelty and Contribution**

We hope that the above details resolve the doubts about our contributions.
- We propose a simple, easy to use architecture, which tailors any transformer to DAGs.
- It is proven effective over a wide range of well-known benchmarks.
- In particular, it makes various baseline transformers we tested competitive to or outperform proven neural networks which were designed for DAGs.
- The qualitative performance is complemented by  considerable gains in efficiency, which is impressive for transformers in general but also in comparison to existing DAG architectures, such as DAGNN.

---

### Comment · Area_Chair_vN1V · 2023-08-13
**Please read and respond to rebuttal**

Dear reviewers,

Please read and respond to the rebuttal as soon as possible.

Thanks

---

> ### Author Response · Authors · 2023-08-17
> **Thank you everyone!**
>
> We thank the AC for starting the discussion.
>
> We sincerely appreciate that all reviewers fully recognized our contributions, took our rebuttal into account, and provided constructive and positive feedback!

---

### Decision · Program_Chairs · 2023-09-21

**Decision:**

Accept (poster)

**Comment:**

Three of the reviewers scored the paper with a "weak accept." Their reasons were that the proposed approach is interesting, new, and addresses an important problem. They also valued the experimental results as convincing. The three weaknesses listed by the only reviewer who tended towards rejecting the paper were essentially all identical and addressed the reachability of nodes of the method and the choice of k. Due to the more detailed reviews of the other reviewers, I recommend accepting the paper. In case the paper gets accepted, I would ask the authors to incorporate the related work mentioned by the reviewers. In addition, I am aware of prior work https://arxiv.org/abs/2004.02596 that uses the depth of a node in paths (the order of nodes) of a DAG as positional encoding in a transformer, so is conceptionally quite similar to the proposed positional encoding. The focus here is on knowledge graphs and approximate query answering, so different enough application and problem but still methodologically related.